# PROTEIN GENERATION WITH EMBEDDING LEARNING FOR MOTIF DIVERSIFICATION

**Kevin Michalewicz**[1,2,3]* **Chen Jin**[1] **Philip Teare**[1] **Tom Diethe**[1] **Mauricio Barahona**[2]

**Barbara Bravi**[2] **Asher Mullokandov**[1]†

[1]Centre for AI, Data Science & Artificial Intelligence, Biopharma R&D, AstraZeneca, UK
[2]Department of Mathematics, Imperial College London, UK
[3]Instituto de Ingeniería Biomédica, Universidad de Buenos Aires, Argentina

## ABSTRACT

A fundamental challenge in protein design is the trade-off between generating structural diversity while preserving motif biological function. Current state-of-the-art methods, such as partial diffusion in RFdiffusion, often fail to resolve this trade-off: small perturbations yield motifs nearly identical to the native structure, whereas larger perturbations violate the geometric constraints necessary for biological function. We introduce Protein Generation with Embedding Learning (PGEL), a general framework that learns high-dimensional embeddings encoding sequence and structural features of a target motif in the representation space of a diffusion model's frozen denoiser, and then enhances motif diversity by introducing controlled perturbations in the embedding space. PGEL is thus able to loosen geometric constraints while satisfying typical design metrics, leading to more diverse yet viable structures. We demonstrate PGEL on ten representative cases, which include a cancer-related transcription factor complex, an antibody-antigen complex, and an enzyme. PGEL achieves greater structural diversity, better designability, and improved self-consistency, as compared to partial diffusion. Our results establish PGEL as a general strategy for embedding-driven protein generation allowing for systematic, viable diversification of functional motifs.

## 1 INTRODUCTION

Designing proteins that achieve precise biological functions while allowing for structural diversity has long been a central goal in computational protein design. Recent advances in structure prediction models like AlphaFold (Jumper et al., 2021; Abramson et al., 2024), RoseTTAFold (Baek et al., 2021), ESMFold (Lin et al., 2023) and Boltz (Wohlwend et al., 2024; Passaro et al., 2025) have revolutionized protein generative models and paved the way for improved diffusion models in protein design. Among them, RFdiffusion (Watson et al., 2023), which results from fine-tuning RoseTTAFold, has shown strong performance in both unconditional and conditional generation.

Yet, targeted local modification remains a challenge. A common approach is *partial diffusion* in RFdiffusion, in which a native or designed structure undergoes only a few denoising steps to induce diversification (Watson et al., 2023; Vázquez Torres et al., 2024). However, this method faces a fundamental diversity-fidelity trade-off: small structural perturbations keep near-native conformations, but lack diversity, while larger perturbations induce excessive geometric drift that disrupts functional features (Lin et al., 2024). Overcoming this limitation requires rethinking how diffusion models can introduce controlled variation while still anchoring designs to essential geometric constraints.

A promising direction comes from recent advances in conditional image generation. Models such as Stable Diffusion and Latent Diffusion Models (LDMs) generate images from noise guided by text prompts (Ho et al., 2020; Rombach et al., 2022). Beyond standard prompting, textual inversion

---

*For correspondence: kmichalewicz@fi.uba.ar
†For correspondence: asher.mullokandov@astrazeneca.com

learns new prompt embeddings to represent unseen visual concepts (Gal et al., 2022; Jin et al., 2024). Once learned, these embeddings can be diversified to generate outputs that preserve the original concept while exploring novel variations. Here we adopt this embedding-centric view in the context of protein generation.

We present Protein Generation with Embedding Learning (PGEL), a general framework representing the first adaptation of textual inversion principles to protein diffusion models. PGEL introduces two key approaches with broad applicability: (1) learning high-dimensional embeddings that capture the sequence and structural characteristics of target protein regions of interest, thus shifting the paradigm from coordinate-space to embedding-space perturbations, and (2) relaxing evolutionary and structural constraints by masking embeddings. Although we present our work here using RFdiffusion's representation space, our method is general and readily adaptable to other protein diffusion models, and can thus leverage the rich representational capacity of pre-trained diffusion models without expensive retraining or fine-tuning.

In this work we formalize a new design task, *motif diversification*: given an experimentally characterized structure with a functional motif embedded in a larger scaffold, we aim to generate alternative motif conformations and poses while keeping the rest of the protein fixed in real space. Diversification is defined at the backbone level, meaning that motif residues are allowed to move and rearrange relative to each other and to the scaffold, but without pre-specifying side-chain identities or chemistry. Instead, we require that downstream sequence design and structure prediction can recover sequences that realize each diversified backbone. Biologically, this corresponds to exploring families of alternative structural realizations of an underlying functional motif (*e.g.*, a binding epitope or active site) that remain compatible with the scaffold, enabling the enhancement of properties such as affinity, specificity, or stability while preserving the overall protein architecture. Some existing approaches address related challenges, but differ in scope and implementation: structure inpainting methods (*e.g.*, masked region generation) fully marginalize a region by masking and regenerating it *de novo*, discarding the specific native geometry (Zhang et al., 2023), whereas flexible backbone loop remodeling in Rosetta (KIC/Next-Generation KIC) samples local conformations under explicit geometric and energetic restraints to achieve high-fidelity but relatively localized exploration (Mandell et al., 2009; Stein & Kortemme, 2013; Leman et al., 2020). Hence these tools do not explicitly target controlled exploration of a *neighborhood* around an existing functional motif while keeping a surrounding scaffold nearly fixed.

We, therefore, compare chiefly to partial diffusion in RFdiffusion, the prevailing stochastic baseline for local variation which has been recently applied in therapeutically relevant design settings, including *de novo* creation of high-affinity peptide binders and venom toxin neutralizers (Vázquez Torres et al., 2024; 2025). Across ten representative scenarios, eight proposed in Watson et al. (2023), with two additional cases involving an antibody bound to Alzheimer's disease's amyloid beta peptide and the adenylate kinase enzyme, PGEL (1000 samples) produces more designable structures in nine cases (motif pLDDT $\geq$ 70, scRMSD $\leq$ 1Å, mRMSD $\leq$ 2Å) than partial diffusion. PGEL also yields more structurally diverse TM-score clusters distinguishable from native, and shows better self-consistency after inverse folding and refolding (meeting mRMSD and pAE thresholds), while maintaining predicted binding affinities comparable to native and exceeding those obtained with partial diffusion. Our results support embedding learning combined with masking as a general, efficient strategy for systematic motif diversification.

## 2 BACKGROUND

**Diffusion models for proteins.** Earlier works adapted Denoising Diffusion Probabilistic Models (DDPMs) to protein design by conditioning on local structural elements or coarse fold constraints (Wu et al., 2024; Anand & Achim, 2022; Trippe et al., 2023; Luo et al., 2022) yet, while encouraging, they produced few sequences that refolded to target backbones. RFdiffusion subsequently emerged as the diffusion approach that reliably yields designable structures and sequences that recover the intended geometry. In RFdiffusion, a highly accurate protein structure prediction method (RoseTTAFold (Baek et al., 2021)) is fine-tuned to undo random perturbations of atomic coordinates introduced via 3D Gaussian noise (*i.e.*, to denoise). RFdiffusion can be constrained to specific binding targets, or symmetry specifications, and once trained it can be viewed as a *frozen denoiser*. RoseTTAFold/AlphaFold-style models (including RFdiffusion) learn so-called *state* and

*pair* embeddings (related to per-residue and residue-residue properties of the protein structure, respectively) and MSA embeddings related to multiple sequence alignment (Jumper et al., 2021).

**Textual inversion.** Gal et al. (2022) builds on LDMs (Rombach et al., 2022), a specific class of DDPMs, to perform textual inversion. In the context of text-to-image models, let $x$ represent an image, $s$ a text prompt, $\epsilon_\theta$ a pre-trained denoising network, and $\varepsilon$ an image encoder. LDMs aim to minimize the following loss:

$$\mathcal{L}_{\text{LDM}} := \mathbb{E}_{z\sim\varepsilon(x),s,\epsilon\sim\mathcal{N}(0,1),t} \left[\|\epsilon - \epsilon_\theta(z_t, t, c_\theta(s))\|_2^2\right] \tag{1}$$

Here, $c_\theta(s)$ represents a pre-trained text encoder that conditions the denoiser $\epsilon_\theta$ based on the text prompt $s$, and $z_t$ is a noised version of the image embedding $z$ at timestep $t$. The goal of textual inversion is to learn a new text embedding $v_*$ corresponding to a particular concept $s_*$ such that it minimizes the LDM loss (equation 1). This means conditioning $\epsilon_\theta$ on $v_*$ so the generated image $\tilde{x}$ closely resembles the original image $x$.

# 3 METHODS

We now present our method, *Protein Generation with Embedding Learning (PGEL)*, and describe how we learn the embedding representation of a motif in Section 3.1. In Section 3.2, we propose an approach to increase motif diversity, and Section 3.3 details the evaluation metrics.

## 3.1 PROTEIN GENERATION WITH EMBEDDING LEARNING (PGEL)

We generalize the notion of textual inversion with LDMs to proteins, treating the structure as analogous to an image, and the sequence as analogous to a text prompt. Let $R_*$ be a region of interest, or *motif*, defined as a continuous or discontinuous set of $L_*$ amino acids within a protein. The motif has structure $x_*$ and sequence $s_*$, where the coordinates of $x_*$ are obtained from an experimental Protein Data Bank (PDB) entry, and the sequence $s_*$ is *masked* when passed as an input to PGEL, *i.e.*, the amino acid range of the motif is specified, but not its exact composition.

PGEL learns a representation of $R_*$ in embedding space, which we denote as $v_*$. The remainder of the protein constitutes the *scaffold*, with structure $x_c$ and sequence $s_c$ of length $L_c$, from which the protein LDM frozen ENCODER computes an embedding representation $v_c$.

The procedure (see Figure 1 and Algorithm 1) starts by building a noised protein structure in which the scaffold coordinates are retained while the motif coordinates are subjected to $T$ rounds of Gaussian noise injection, following Trippe et al. (2023). At each timestep $t$, the protein LDM frozen DENOISER predicts a denoised motif structure $\hat{x}_*^{(0)}$, conditioned jointly on the learnable motif embedding $v_*$ and the fixed embedding $v_c$. These embeddings include state, pair and MSA embeddings. Then, by using structure $x^{(t)}$ and the intermediate structure $[x_c, \hat{x}_*^{(0)}]$, a reverse diffusion step RE-VERSESTEP, which does not contain any learnable parameters, yields $x^{(t-1)}$ (see Algorithm 3). In practice, we employ pre-trained building blocks of RFdiffusion for both the ENCODER and DE-NOISER, though alternative models could be substituted if desired.

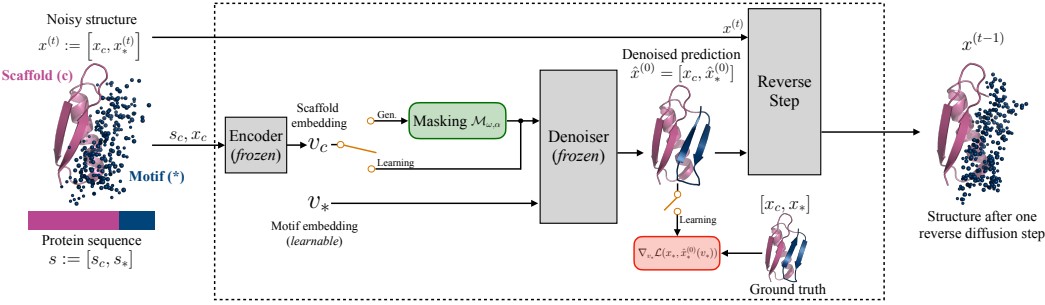

Figure 1: Outline of the PGEL learning and generation procedures during one reverse diffusion step.

**Embedding optimization.** The embedding $v_*$ is learned by minimizing:

$$\mathcal{L} = \mathcal{L}_{\text{MSE}} + \lambda_{\text{DM}}\mathcal{L}_{\text{DM}} + \lambda_{\text{torsion}}\mathcal{L}_{\text{torsion}} \tag{2}$$

This loss function is composed of three terms, described hereafter, which compare different features of the ground truth structure $x_*$ and the predicted structure $\hat{x}_*^{(0)}(v_*)$ of the motif with the coefficients $\lambda_{\text{DM}}, \lambda_{\text{torsion}} \in \mathbb{R}_{\geq 0}$ controlling the relative weight of the terms.

**Data fidelity term (backbone atoms).** For each motif residue $i \in R_*$ we consider the $A = 4$ backbone atoms (nitrogen N, $\alpha$-carbon $C_\alpha$, carbon C, oxygen O). Let $\hat{x}_{i,a}^{(0)} \in \mathbb{R}^3$ denote the predicted position of atom $a$ in residue $i$ and $x_{i,a} \in \mathbb{R}^3$ its ground truth counterpart. We then compute the mean squared error (MSE) between the backbone atoms of the ground truth and predicted motif:

$$\mathcal{L}_{\text{MSE}}(x_*, \hat{x}_*^{(0)}(v_*)) = \frac{1}{AL_*} \sum_{i \in R_*} \sum_{a \in \{\text{N},\text{C}_\alpha,\text{C},\text{O}\}} \left\| \hat{x}_{i,a}^{(0)}(v_*) - x_{i,a} \right\|^2 \tag{3}$$

**Distance matrix between $\alpha$-carbons.** Let $\hat{x}_{i,\text{C}_\alpha}^{(0)} \in \mathbb{R}^3$ denote the predicted position of the $\alpha$-carbon atom in residue $i$, and $x_{i,\text{C}_\alpha} \in \mathbb{R}^3$ its ground truth counterpart. We define the following loss term based on $\alpha$-carbon Distance Matrices (DM), inspired by the distrogram notion (Senior et al., 2020):

$$\mathcal{L}_{\text{DM}}(x_*, \hat{x}_*^{(0)}(v_*)) = \frac{1}{L_*^2} \sum_{i \in R_*} \sum_{j \in R_*} \left( \left\| \hat{x}_{i,\text{C}_\alpha}^{(0)}(v_*) - \hat{x}_{j,\text{C}_\alpha}^{(0)}(v_*) \right\| - \left\| x_{i,\text{C}_\alpha} - x_{j,\text{C}_\alpha} \right\| \right)^2 \tag{4}$$

In contrast to $\mathcal{L}_{\text{MSE}}$, $\mathcal{L}_{\text{DM}}$ is invariant under rigid motions (translations and rotations), thus encouraging global shape consistency.

**Backbone torsion angles.** Let $\hat{\phi}_i$ and $\hat{\psi}_i$ denote the predicted backbone torsion angles at residue $i$, computed from $\hat{x}_*^{(0)}(v_*)$, and let $\phi_i$ and $\psi_i$ be the corresponding ground truth values (Ramachandran et al., 1963). We impose a constraint on angular torsions through a cosine-based loss term akin to that of AlphaFold (Jumper et al., 2021):

$$\mathcal{L}_{\text{torsion}}(x_*, \hat{x}_*^{(0)}(v_*)) = \frac{1}{L_* - 2} \sum_{i=2}^{L_* - 1} \left[ 1 - \cos\left( \hat{\phi}_i(v_*) - \phi_i \right) + 1 - \cos\left( \hat{\psi}_i(v_*) - \psi_i \right) \right] \tag{5}$$

This term penalizes sterically implausible geometries, helping improve performance under the predicted local distance difference test (pLDDT). We do not include the third backbone angle $\omega$ as it is typically considered fixed at 180 degrees (Cutello et al., 2006).

---

**Algorithm 1** PGEL – Embedding learning

---

**Input:** region of interest/motif $R_*$ with masked sequence $s_*$ and structure $x_*$, fixed scaffold with sequence $s_c$ and structure $x_c$, pre-trained ENCODER and DENOISER.
**Output:** learned embedding $v_*$ for region $R_*$.
initialize $v_*$ with zeros.
**while** not converged **do**
    Build noised structure $x^{(T)} := [x_c, x_*^{(T)}]$ with associated sequence $s := [s_c, s_*]$.
    $v_c = \text{ENCODER}(s_c, x_c)$
    **for** $t = T$ **down to** $1$ **do**
        $\hat{x}_*^{(0)} = \text{DENOISER}(v_c, v_*)$
        $x^{(t-1)} = \text{REVERSESTEP}(x^{(t)}, [x_c, \hat{x}_*^{(0)}])$
        Update $v_*$ by taking a gradient step $\nabla_{v_*} \mathcal{L}(x_*, \hat{x}_*^{(0)}(v_*))$
    **end for**
**end while**
**Return** $v_*$

---

Once the embeddings are learned for a particular protein, we employ Algorithm 2 to generate novel proteins with diversified region of interest $R_*$ (see Figure 1). We demonstrate the necessity of learning motif embeddings by comparing with simpler baselines in Appendix D.

## 3.2 ENHANCING THE DIVERSITY OF GENERATED MOTIFS

MSA embeddings in sequence-to-structure predictors contain evolutionary covariation information about residues, thereby capturing geometric constraints such as residue proximity. In RFdiffusion,

---

**Algorithm 2** PGEL – Generation with embedding masking

---

**Input:** region of interest/motif $R_*$ with learned embeddings $v_*$, fixed scaffold with sequence $s_c$ and structure $x_c$, pre-trained ENCODER and DENOISER.
**Output:** generated structure.

Build noised structure $x^{(T)} := [x_c, x_*^{(T)}]$.
Draw at random the sample masking type $\omega \sim \mathrm{Ber}(\frac{1}{2})$ (row if 0, column if 1).
Sample masking rate $\alpha \sim \mathcal{U}[0,1]$.
Define $\mathcal{M}_{\omega,\alpha}(\cdot)$ as a zero mask with type $\omega$ and rate $\alpha$.
$v_c = \mathrm{ENCODER}(s_c, x_c)$
**for** $t = T$ **down to** 1 **do**
$\quad \hat{x}_*^{(0)} = \mathrm{DENOISER}(\mathcal{M}_{\omega,\alpha}(v_c), v_*)$
$\quad x^{(t-1)} = \mathrm{REVERSESTEP}(x^{(t)}, [x_c, \hat{x}_*^{(0)}])$
**end for**
**Return** $x^{(0)}$

---

however, such embeddings are derived solely from the input sequence $s := [s_c, s_*]$ rather than from a full stack of aligned sequences, and can be represented as a $d_{\mathrm{MSA}} \times L$ matrix, where $d_{\mathrm{MSA}}$ is the depth of the MSA embeddings and $L := L_* + L_c$ the total protein length. With PGEL, we show that the diversity of generated structures can be increased by applying perturbations to the scaffold MSA embeddings $v_c^{\mathrm{MSA}} \in \mathbb{R}^{d_{\mathrm{MSA}} \times L_c}$. These embeddings couple through attention mechanisms with state and pair embeddings produced by an internal RFdiffusion encoder, and also interact with the learned motif embedding $v_*$, which provides an independent conditioning signal for the frozen denoiser (see Appendix A for more details).

**Embedding masking.** We studied the effect of applying zero masks, *i.e.* masks zeroing specific elements, to the scaffold MSA embeddings during generation (see Algorithm 2). *Row masking* corresponds to masking specific features for all residues, whereas *column masking* zeroes out all features of specific residues. Both strategies lift some constraints on inter-residue distances, and modulate which co-variation patterns remain accessible. We sample $\omega \sim \mathrm{Ber}(\frac{1}{2})$ to choose the masking mode ($\omega = 0$ for row masking and $\omega = 1$ for column masking) and $\alpha \sim \mathcal{U}[0,1]$, the masking rate, to set the fraction of rows or columns masked. This defines the operator $\mathcal{M}_{\omega,\alpha}(\cdot)$, which implements zero masking with type $\omega$ and rate $\alpha$. As such, masking $v_c$ relaxes the geometric constraints of the generated motif (see Appendix B), while its omission results in no diversity, as shown in Appendix D. We also assessed the correlation between motif structural diversity and rate $\alpha$ in Appendix E.

## 3.3 EVALUATION METRICS

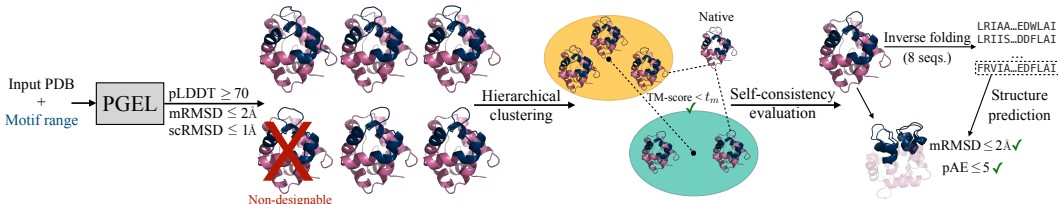

Figure 2: Summary of the evaluation metrics. PGEL takes as input a PDB entry and the amino acid range corresponding to the motif. From 1000 PGEL-generated backbones, designable candidates are filtered by root mean square deviation (RMSD) and pLDDT thresholds, and structural diversity is assessed via hierarchical clustering. Backbones are also required to be *distinguishable* from the native. Cluster representatives undergo self-consistency evaluation: sequences assigned to the designable backbones with ProteinMPNN are refolded, and at least one predicted structure must satisfy set mRMSD and predicted alignment error (pAE) conditions relative to the generated backbone.

**Designability.** To quantify designability in the motif diversification task, we first require a motif $\mathrm{pLDDT} \geq 70$ as computed by an RFdiffusion internal block, following the threshold adopted for

related tasks by Lin et al. (2024). We also require the scaffold RMSD to be $\mathrm{scRMSD} \leq 1\text{Å}$, to ensure that the residues surrounding the motif remain fixed. Finally, we set the threshold for the motif RMSD to $\mathrm{mRMSD} \leq 2\text{Å}$, to allow for structural diversification of the motif.

**Diversity.** To quantify structural diversity among generated proteins (Figure 2), we compute the pairwise TM-scores (Zhang & Skolnick, 2004) across all designable candidates, and employ hierarchical clustering (Lin et al., 2024) with several linkage thresholds $t_m$ to group similar backbones under this score. Diversity is measured by the number of clusters. We evaluate also the TM-score with respect to the native motif: a cluster is considered *distinguishable* from native if, for TM-score threshold $t_m \in [0, 1]$, at least one cluster member exhibits lower similarity than $t_m$ relative to the native. This analysis ensures that we capture the structural distinctiveness of the backbones.

**Self-consistency.** For the distinguishable backbones we use the procedure in Trippe et al. (2023) based on inverse folding to assess self-consistency between generated and predicted structures. Specifically, we use ProteinMPNN with default parameters (Dauparas et al., 2022) to assign 8 plausible sequences to each backbone, followed by a sequence-to-structure model, here AlphaFold3 (Abramson et al., 2024), to predict 8 full proteins. A designed backbone is deemed self-consistent if it satisfies for at least one of the 8 predicted structures: $\mathrm{mRMSD} \leq 2\text{Å}$ and $\mathrm{pAE} \leq 5$ (Figure 2). Previous studies included this procedure under the designability assessment (Watson et al., 2023; Lin et al., 2024). However, this is computationally expensive and, when prioritizing diversity, often inefficient: many backbones either fail the initial scRMSD or mRMSD filters or exhibit negligible structural diversity. In motif diversification, diversity among generated proteins and distinguishability with respect to the native are decisive. We therefore invert the pipeline to enforce these criteria first and reserve the costly self-consistency evaluation only for diverse candidates.

**Binding affinity.** For protein-protein complexes, we run PRODIGY (Vangone & Bonvin, 2015; Xue et al., 2016) to estimate the binding affinity $\Delta G$ expressed in kcal/mol, with larger $|\Delta G|$ values indicating stronger binding. It is desirable that new designs present binding affinity values comparable to, or larger in magnitude than, those of the native complex. Note that learning is not optimized to enhance binding affinity; rather, this serves as an *a posteriori* assessment.

## 4 EXPERIMENTS

We focus on ten representative test cases. Eight were proposed for different tasks by Watson et al. (2023), and two were added in this work: a TAP01 family antibody in complex with an amyloid beta peptide, which is related to Alzheimer's disease (van Dyck, 2018), and a mutated version of the adenylate kinase enzyme. We present the results for all cases in Figure 3 and Tables 3 and 4, and we describe in detail the cases of a monomer and a p53-MDM2 complex.

### 4.1 PROTOCOL

We established a protocol to systematically compare our method with RFdiffusion's partial diffusion using the metrics introduced in Section 3.3. For partial diffusion, we generated 1000 protein backbones by uniformly sampling the number of diffusion timesteps, $T \sim \mathcal{U}\{2, 3, \ldots, 49\}$, as $T = 50$ corresponds to the full diffusion process in RFdiffusion. In this way, we cover a spectrum of structural perturbations ranging from near-native backbones to unrelated ones. For PGEL, we performed the learning of $v_*$ with Stochastic Gradient Descent with learning rate $l_r = 4 \times 10^{-4}$ and momentum $p = 0.9$, $\lambda_{\mathrm{DM}} = 0.01$ and $\lambda_{\mathrm{torsion}} = 0.05$ (Algorithm 1), and we then generated 1000 protein backbones (Algorithm 2).

For both sets of 1000 generated structures, we evaluated designability and, among those deemed designable, we computed TM-scores between all generated motifs and with respect to the native structure. We then plotted the number of clusters as a function of the TM-score. For structures that were designable and diverse according to a typical TM-score threshold $t_m = 0.6$ (Lin et al., 2024), we performed inverse folding through ProteinMPNN to generate compatible sequences, followed by AlphaFold3 inference to assess whether the predicted sequences refolded into the intended backbones, fulfilling the self-consistency requirement defined in Section 3.3.

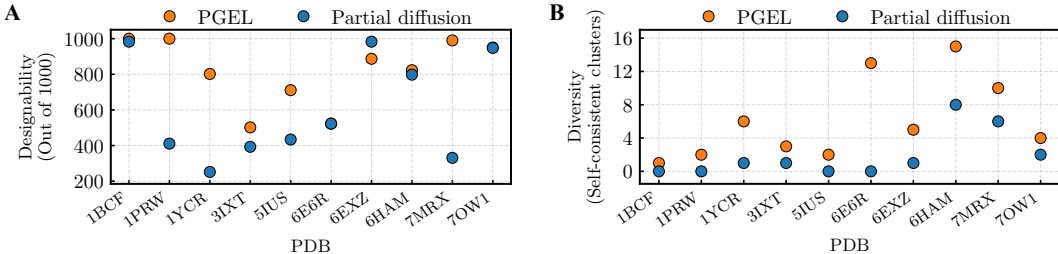

Figure 3: Comparison of partial diffusion in RFdiffusion and PGEL. (A) Number of viable structures out of 1000. (B) Number of self-consistent clusters (diversity), computed at TM-score threshold $t_m = 0.6$.

## 4.2 EXAMPLE 1: MONOMER

Calmodulin (PDB entry: 1PRW), a monomeric protein containing a double EF-hand motif spanning residues 16-35 and 52-71 (structured multi-motif), was considered as the representative test case for single-chain proteins. All the 1000 backbones candidates generated by PGEL resulted to be designable, well exceeding the 411 obtained by partial diffusion (Figure 3A, Table 3). All of the backbones generated by partial diffusion satisfied the pLDDT constraint, consistent with the fact that RFdiffusion's training favors high-confidence local structures, but 589 of them failed to meet the expected motif RMSD threshold. In these cases, the added noise during diffusion excessively perturbed the initial backbone, leading to conformations that no longer preserved the intended geometry of the EF-hand motif.

We then evaluated the structural diversity of the designable backbones, recording the number of clusters as a function of the TM-score threshold (Figure 4A). PGEL consistently produced a higher number of clusters across thresholds, demonstrating that embedding perturbations through masking introduce greater variability in backbone conformations. On the other hand, partial diffusion yielded structures too similar to the native backbone, and hence not distinguishable from it.

We carried out the self-consistency assessment at $t_m = 0.6$, as per protocol. For PGEL, the two clusters had backbones that successfully refolded into the intended conformations after sequence design and AlphaFold3 inference. Figure 4B illustrates these two successful cases, along with examples of backbones generated by partial diffusion that either did not satisfy the mRMSD metric condition or the distinguishability from native.

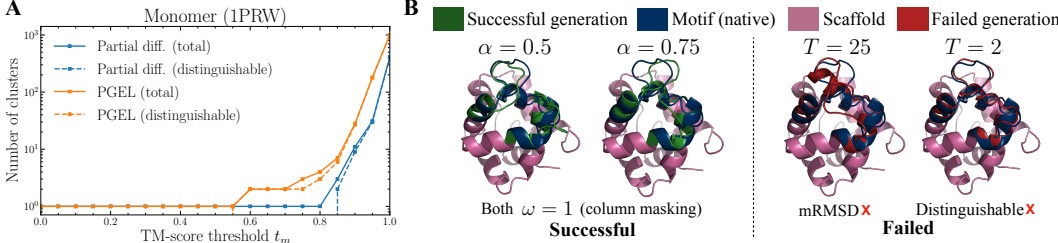

Figure 4: Results for a monomer (PDB: 1PRW). (A) Number of clusters, both total and distinguishable from native, as a function of the TM-score threshold $t_m$ for PGEL and partial diffusion. (B) *Left:* two successful PGEL designs at $t_m = 0.6$ using column masking with rates $\alpha = 0.5$ and $\alpha = 0.75$. *Right:* two partial diffusion failed backbones at $t_m = 0.6$, one obtained with $T = 25$ timesteps that violates the motif RMSD constraint, and one with $T = 2$ timesteps that is not distinguishable from the native.

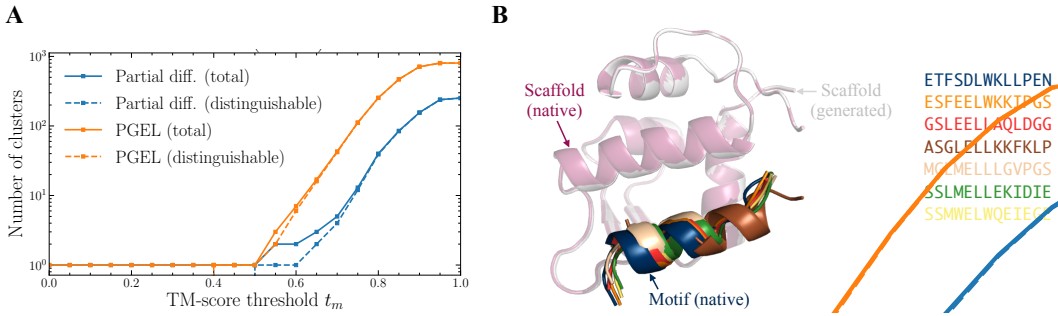

Figure 5: Results for a p53-MDM2 complex (PDB: 1YCR). (A) Number of clusters identified by PGEL and partial diffusion, both total and distinguishable from native, as a function of the TM-score threshold $t_m$. (B) Generated binding site backbones (overlaid with native), alongside the native sequence and sequences that refold to self-consistent structures.

### 4.3 EXAMPLE 2: COMPLEX

The interaction between the transcription factor p53 and its negative regulator MDM2 is a key molecular process in cancer progression. Specifically, pharmacological disruption of the p53-MDM2 complex restores p53 activity and has been proven beneficial in cancer therapy (Hu et al., 2021).

Starting from PDB entry 1YCR, we addressed the motif diversification task by redesigning the complete p53 under the RMSD constraints described in Section 3.3. PGEL generated 802 designable backbones out of 1000 trials, with most non-designable cases attributable to low pLDDT confidence scores (Table 3). Partial diffusion produced only 252 designable backbones with considerably reduced structural diversity (a single cluster at TM-score threshold $t_m = 0.6$, see Figure 5B). PGEL, by comparison, gave six clusters at $t_m = 0.6$, all of which passed the self-consistency checks.

When assessing the binding affinity *a posteriori*, five out of six representatives of PGEL clusters exhibited lower affinity compared to the sole valid instance of partial diffusion (Table 5). Notably, sequence SSMWELWQEIEGE (see Figure 5B), designed with PGEL in combination with Protein-MPNN, folded, as predicted by AlphaFold3, into a structure with a binding affinity comparable to that of the native structure, despite sharing only around 15% of sequence identity. This result highlights PGEL's ability to generate backbones that can accommodate sequences unrelated to the native while refolding into structures that preserve function.

**Computational remarks.** Across the ten case studies, column masking was more effective than row masking (see Table 4, with over 60% of successful outcomes with $\omega = 1$). The time required per timestep during the generation process is nearly indistinguishable between PGEL and partial diffusion: average timestep $0.7\,s$ for partial diffusion and $0.71\,s$ for PGEL on a single NVIDIA GeForce RTX 3090 GPU with 24GB of memory.

## 5 LIMITATIONS AND FUTURE WORK

PGEL inherits the biases and limitations of the underlying frozen RFdiffusion denoiser, including its training data distribution and architectural constraints. Moreover, learning embeddings requires additional optimization time, which ranged in our examples from 2 (PDB: 7MRX) to 20 minutes (PDB: 1YCR) on a single GPU, with a trade-off between speed and improved results (more details in Appendix F). Our experiments were limited to motifs of up to 40 residues, with practical limits of around 50 residues given available memory, though scaling to longer motifs should be feasible with larger hardware or engineering optimization. Beyond this, our evaluation is entirely *in silico* (pLDDT/RMSD/TM-score filtering, AlphaFold3 refolding, and PRODIGY $\Delta G$) and thus predictive rather than experimental.

Future work will investigate alternative ways of perturbing embeddings, as this strategy for motif diversification remains largely unexplored, as well as different strategies for sampling the masking parameters $\omega$ and $\alpha$. For instance, instead of sampling $\alpha$ uniformly between 0 and 1, one could bias

it toward smaller masking rates (*e.g.*, using a Poisson distribution with rate $\lambda$, where $\lambda$ tunes how conservative or aggressive the masking is), thus providing finer control over structural perturbations. A more systematic mapping between PGEL's $\omega$ and $\alpha$ and partial diffusion's $T$ would also clarify the relationship between the diversity-fidelity trade-off in both methods. Finally, experimental validation will be pursued in follow-up work.

**Code availability.** Upon publication, we will release code and configurations at `github.com/AstraZeneca/PGEL` to facilitate reproducibility.

## MEANINGFULNESS STATEMENT

Meaningful representations of life, in our context, should capture biologically significant structure/sequence in a way that enables understanding and manipulation. PGEL is a framework that learns representations encoding the sequence and structure of functional motifs–critical components of life–within a diffusion model's latent space. The motif representation is optimized to capture meaningful geometry (distances, torsions), is explicit and separable from the fixed scaffold, and can thus be perturbed in a principled and interpretable way: our MSA masking procedure controllably removes evolutionary information and can reveal which specific features enforce rigid geometric constraints versus those allowing structural flexibility.

## AUTHOR CONTRIBUTIONS

Study concept and design: K.M., C.J., A.M.; Development of source code: K.M.; Analysis and interpretation of data: K.M., A.M.; Writing and revision of the manuscript: K.M., C.J., P.T., T.D., M.B., B.B. and A.M.; Study supervision: C.J., M.B., B.B. and A.M.

## ACKNOWLEDGMENTS

K.M. acknowledges support from the President's PhD Scholarship at Imperial College London. M.B. acknowledges support by the Engineering and Physical Sciences Research Council (EPSRC) under grant EP/N014529/1 funding the EPSRC Centre for Mathematics of Precision Healthcare at Imperial College London. All authors are grateful to Anshul Kanakia, Dino Oglic, Lorena Roldán Martín and Talip Uçar for helpful discussions.

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

## A  REPRESENTATION AND CONDITIONING ARCHITECTURE

RFdiffusion, like RoseTTAFold, maintains three representation "tracks": an MSA track $v^{\mathrm{MSA}}$, a pair track $v^{\mathrm{pair}}$, and a per-residue state track $v^{\mathrm{state}}$ (Baek et al., 2021; Watson et al., 2023). For a protein of length $L$, we write these as

$$v^{\mathrm{MSA}} \in \mathbb{R}^{d_{\mathrm{MSA}} \times L}, \quad v^{\mathrm{pair}} \in \mathbb{R}^{d_{\mathrm{pair}} \times L \times L}, \quad v^{\mathrm{state}} \in \mathbb{R}^{d_{\mathrm{state}} \times L}. \tag{6}$$

In all experiments we instantiate PGEL's denoiser on the released RFdiffusion "Base" model (Base_ckpt.pt), whose public configuration specifies $d_{\mathrm{state}} = 16$, $d_{\mathrm{MSA}} = 256$ and $d_{\mathrm{pair}} = 128$; we keep all model weights and these dimensions fixed.

We partition the sequence positions into motif residues $R_*$ with $|R_*| = L_*$ and scaffold residues $R_c$ with $|R_c| = L_c$, so that $L = L_c + L_*$. Running the frozen RFdiffusion encoder on the scaffold with sequence $s_c$ and structure $x_c$ yields scaffold embeddings

$$v_c^{\mathrm{MSA}} \in \mathbb{R}^{d_{\mathrm{MSA}} \times L_c}, \quad v_c^{\mathrm{pair}} \in \mathbb{R}^{d_{\mathrm{pair}} \times L_c \times L_c}, \quad v_c^{\mathrm{state}} \in \mathbb{R}^{d_{\mathrm{state}} \times L_c}. \tag{7}$$

PGEL introduces learnable motif embeddings

$$v_* = \left( v_*^{\mathrm{MSA}}, \, v_*^{\mathrm{pair}}, \, v_*^{\mathrm{state}} \right), \tag{8}$$

where $v_*^{\mathrm{MSA}} \in \mathbb{R}^{d_{\mathrm{MSA}} \times L_*}$ and $v_*^{\mathrm{state}} \in \mathbb{R}^{d_{\mathrm{state}} \times L_*}$. $v_*^{\mathrm{pair}} \in \mathbb{R}^{d_{\mathrm{pair}} \times L \times L_*}$ collects all pairwise features involving at least one motif residue (motif-motif and motif-scaffold blocks). The only trainable parameters in PGEL are the entries of $v_*$; all RFdiffusion parameters are frozen.

At each optimization step, we construct composite tracks that align with the RFdiffusion denoiser inputs by combining scaffold and motif embeddings. For the MSA and state tracks we concatenate along the sequence dimension:

$$v^{\text{MSA}} = \text{concat}\big(v_c^{\text{MSA}}, v_*^{\text{MSA}}\big) \in \mathbb{R}^{d_{\text{MSA}} \times L}, \quad v^{\text{state}} = \text{concat}\big(v_c^{\text{state}}, v_*^{\text{state}}\big) \in \mathbb{R}^{d_{\text{state}} \times L}. \quad (9)$$

The full pair tensor $v^{\text{pair}} \in \mathbb{R}^{d_{\text{pair}} \times L \times L}$ is obtained by keeping the scaffold-scaffold block equal to $v_c^{\text{pair}}$ and inserting $v_*^{\text{pair}}$ into all rows/columns corresponding to motif positions. The frozen RFdiffusion denoiser then takes the noisy coordinates $x(t)$ together with $\big(v^{\text{MSA}}, v^{\text{pair}}, v^{\text{state}}\big)$ and produces a prediction $\hat{x}^{(0)}(v_*)$. We use the motif slice $\hat{x}_*^{(0)}(v_*)$ in the PGEL loss, while the scaffold coordinates are clamped to $x_c$ when applying the reverse diffusion update from $x(t)$ to $x(t-1)$.

Information exchange between the learned motif embedding and the fixed scaffold embedding occurs entirely through the existing RFdiffusion blocks: global self-attention in the MSA track, MSA-pair and pair-state couplings, and the $SE(3)$-equivariant refinement network. Gradients from the structural and torsion losses propagate through these layers to update $v_*$, while $v_c$ and all network weights remain fixed.

## B  Masking

Multiple-sequence-alignment (MSA) based structure predictors, such as AlphaFold, RoseTTAFold and RFdiffusion, derive a set of MSA embeddings that encode evolutionary covariation between residue positions and therefore impose strong geometric constraints on the predicted structure. In RFdiffusion these embeddings can be represented as a matrix $v_c \in \mathbb{R}^{d_{\text{MSA}} \times L_c}$, where $d_{\text{MSA}}$ is the MSA embedding depth and $L_c$ is the number of scaffold residues. Motivated by recent work that uses MSA column masking in AlphaFold-based pipelines to weaken coevolutionary constraints and increase conformational diversity (Kalakoti & Wallner, 2025; Piomponi et al., 2025), we introduce a simple masking operator $\mathcal{M}_{\omega, \alpha}(\cdot)$ that perturbs the scaffold MSA embedding during PGEL generation.

Specifically, given a masking type $\omega \sim \text{Ber}(\frac{1}{2})$ and a masking rate $\alpha \sim \mathcal{U}[0, 1]$ we construct a binary mask on $v_c$:

- Row masking ($\omega = 0$) selects a fraction $\alpha$ of rows of $v_c$ (MSA embedding features) and sets them to zero for all scaffold residues. This reduces the effective dimensionality of the MSA feature space while preserving which residues are constrained.

- Column masking ($\omega = 1$) selects a fraction $\alpha$ of columns of $v_c$ (scaffold residue positions) and sets all their MSA features to zero. This locally removes the evolutionary context of specific scaffold residues while leaving the remaining scaffold positions fully constrained.

In both cases, masking weakens the coevolutionary signals that couple the scaffold to the motif and thereby relaxes the induced geometric constraints. In our implementation, masking is applied only to the scaffold MSA embedding: the scaffold state and pair embeddings, as well as the learned motif embedding $v_*$, are left unmodified so any changes in the motif under different masks are mediated by the way EvoFormer attention blocks mix MSA, state, and pair embeddings. The frozen RFdiffusion denoiser therefore receives two conditioning signals at each reverse step: the perturbed scaffold MSA embedding $\mathcal{M}_{\omega, \alpha}(v_c)$ and the learned motif embedding $v_*$.

Empirically, we observe that column masking ($\omega = 1$) is substantially more effective than row masking in generating diverse yet designable motifs (around $80\%$ of successful generations in Table 4 employ column masking). This is consistent with the interpretation above: while row masking removes global features that are shared across all residues and therefore tends to preserve the relative pattern of constraints, zeroing entire columns selectively weakens constraints arising from a subset of scaffold residues that are strongly coevolving with the motif and which enforce the native motif geometry. (This local relaxation allows the motif to move relative to the scaffold frame, while the rest of the scaffold remains tightly constrained by the unmasked columns). An ablation where we disable masking altogether ($\mathcal{M}_{\omega, \alpha} \equiv \text{Id}$) shows that, while PGEL still produces designable structures, the diversity collapses to a single TM-score cluster per system at threshold $t_m = 0.6$,

confirming that masking is necessary to move beyond the native motif basin while maintaining designability (Appendix D).

## C  REVERSESTEP ALGORITHM

Let $x^{(t)} = \{(r_l^{(t)}, u_l^{(t)})\}_{l=1}^L$ denote the noisy protein backbone structure at diffusion step $t$, where each residue $l$ is represented by a rotation $r_l^{(t)} \in SO(3)$, with $SO(3)$ the special orthogonal group in three dimensions, and a translation $u_l^{(t)} \in \mathbb{R}^3$. Let $\hat{x}^{(0)} = \{(\hat{r}_l^{(0)}, \hat{u}_l^{(0)})\}_{l=1}^L$ denote the predicted denoised structure. Let $\{\beta^{(t)}\}_{t=1}^T$ be a variance schedule with $\gamma^{(t)} = 1 - \beta^{(t)}$ and $\bar{\gamma}^{(t)} = \prod_{s=1}^t \gamma^{(s)}$. For translations, let $u_l^{(t-1)}$ be sampled from a Gaussian distribution with covariance $\beta^{(t)} I_3$. For rotations, let $s_l$ denote the score approximation presented in Watson et al. (2023), $\epsilon_{l,d}$ isotropic Gaussian perturbations and $\{f_d\}_{d=1}^3$ a basis of the Lie algebra $SO(3)$.

---

**Algorithm 3** REVERSESTEP function (Watson et al., 2023)

---

**Input:** noisy structure $x^{(t)}$, denoised prediction $\hat{x}^{(0)}$.
**Output:** updated structure $x^{(t-1)}$.
**for** $l = 1, \dots, L$ **do**
  $(r_l^{(t)}, u_l^{(t)}) = x_l^{(t)}$
  $(\hat{r}_l^{(0)}, \hat{u}_l^{(0)}) = \hat{x}_l^{(0)}$
  $u_l^{(t-1)} \sim \mathcal{N}\left( \frac{\sqrt{\bar{\gamma}^{(t-1)}}\beta^{(t)}}{1-\bar{\gamma}^{(t)}}\hat{u}_l^{(0)} + \frac{\sqrt{\gamma^{(t)}}(1-\bar{\gamma}^{(t-1)})}{1-\bar{\gamma}^{(t)}}u_l^{(t)}, \; \beta^{(t)} I_3 \right)$
  // Updating rotations below
  $s_l = \text{ROTATIONSCOREAPPROXIMATION}(r_l^{(t)}, \hat{r}_l^{(0)}, \sigma_t^2)$
  $\epsilon_{l,1}, \epsilon_{l,2}, \epsilon_{l,3} \overset{\text{iid}}{\sim} \mathcal{N}(0,1)$
  $r_l^{(t-1)} = r_l^{(t)} \exp_{I_3}\left\{ (\sigma_t^2 - \sigma_{t-1}^2)r_l^{(t)\top}s_l + \sqrt{\sigma_t^2 - \sigma_{t-1}^2}\sum_{d=1}^3 \epsilon_{l,d}f_d \right\}$
  $x_l^{(t-1)} = (r_l^{(t-1)}, u_l^{(t-1)})$
**end for**
**Return** $x^{(t-1)}$

---

## D  BASELINES

To demonstrate the necessity of both learned embeddings and zero masking, we consider simpler baselines in which one of these steps is omitted.

We first show that the presence of a fix scaffold, which informs the frozen denoiser via computed scaffold embeddings $v_c$ (see Algorithm 1 and Appendix A), is insufficient to produce constraint-satisfying structures if the motif embeddings $v_*$ are not learned. To test this, we skip Algorithm 1 and proceed directly to the generation with embedding masking (Algorithm 2). At each timestep, we now obtain the predicted motif structure $\hat{x}_*^{(0)}$ employing a masked version of the scaffold embeddings $\mathcal{M}_{\omega,\alpha}(v_c)$ and with $v_*$ either set to zero or to random values drawn from a standard normal distribution. In both scenarios, we observe a total of 0 successful generations across all experiments. For zero embeddings, we obtain a mean mRMSD of 3.11Å and a mean pLDDT of 49 across the ten examples, while these values are 3.42Å and 44, respectively, for random embeddings.

It could be argued that the stochasticity of RFdiffusion's denoiser, even with frozen parameters, is sufficient to generate diversified motifs after embedding learning. Nevertheless, when computing at each timestep the predicted motif structure as $\hat{x}_*^{(0)} = \text{DENOISER}(v_c, v_*)$, only a single self-consistent cluster is obtained at $t_m = 0.6$ per experiment.

# E  MASKING RATE AND ITS IMPACT ON STRUCTURAL DIVERSITY

We studied the effect of the masking rate $\alpha \in [0, 1]$ on the resulting structural diversity, quantified by the TM-score between each generated motif and the native structure. We plotted $1 - \text{TM-score}$ as a function of $\alpha$ for all the experiments, obtaining moderate correlations as measured by the Pearson correlation score $r$ with statistical significance (Figure 6).

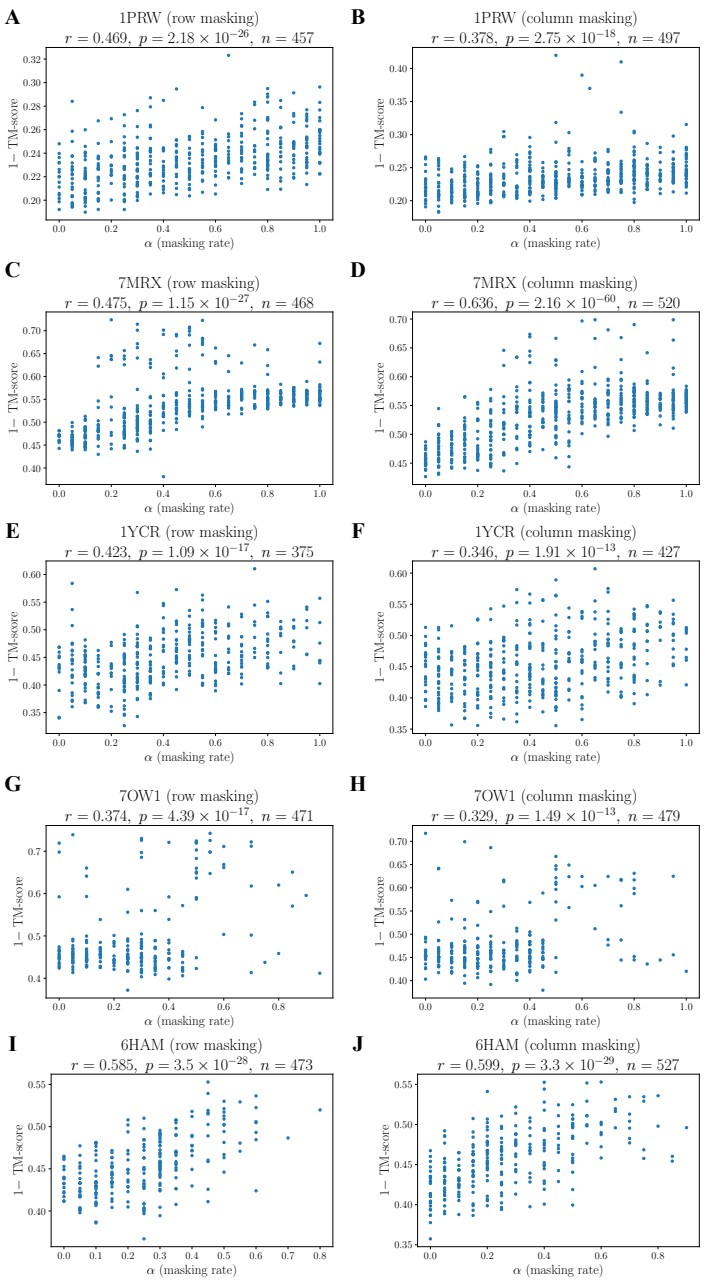

Figure 6: $1 - \text{TM-score}$ of each generated structure deemed designable (with respect to the native motif) as a function of the masking rate $\alpha$, for row and column masking and five examples. For each case, we report the Pearson correlation coefficient $r$, its associated $p$-value, and the sample size.

## F   COMPUTATIONAL PERFORMANCE: STANDALONE *vs.* LINKED MOTIFS

We observed that the learning process described in Algorithm 1 requires more time when the designed motif is a standalone protein, *i.e.*, a full chain. The results indicate that the computational performance appears largely independent of the motif length $L_*$ (Table 1).

Table 1: Learning time comparison for standalone *vs.* linked motif designs.

| PDB | Standalone | $L_*$ | Learning time (min.) |
|-----|:----------:|:-----:|:--------------------:|
| 1PRW | × | 42 | 3.20 |
| 7MRX | × | 22 | 2.06 |
| 7OW1 | × | 8 | 4.18 |
| 6HAM | × | 22 | 2.48 |
| 1YCR | ✓ | 13 | 19.63 |
| 7WT5 | ✓ | 8 | 34.87 |
| 3MZW | ✓ | 58 | 23.41 |
| 1ATP | ✓ | 20 | 32.12 |

## G   STUDY OF REGULARIZATION HYPERPARAMETERS

We studied which combinations of $\lambda_{\mathrm{DM}}$ and $\lambda_{\mathrm{torsion}}$ minimize the mRMSD at convergence of Algorithm 1 (see equation 2). Further, we examined the contribution of individual loss terms $\mathcal{L}_{\mathrm{DM}}$ and $\mathcal{L}_{\mathrm{torsion}}$ by performing ablation experiments in which either the distance matrix term (equation 4) or the backbone torsion angle term (equation 5) were removed.

Although all combinations in Table 2 satisfy the mRMSD $< 2$Å constraint, the configurations with $\lambda_{\mathrm{torsion}} = 0.05$ and $\lambda_{\mathrm{DM}} \in \{0.01, 0.05, 0.1\}$, as well as $\lambda_{\mathrm{torsion}} = \lambda_{\mathrm{DM}} = 0.1$ appear to provide the best performance. While these results offer some guidance on the appropriate orders of magnitude of the regularization strengths, the full validation pipeline should be run for each case to assess the number of successes.

Regarding the ablation study, in the three examples the pLDDT values are no longer constraint-satisfying at convergence, with a mean value of $65$ across the three experiments, when $\mathcal{L}_{\mathrm{torsion}}$ is set to zero. When $\mathcal{L}_{\mathrm{DM}} = 0$, mRMSD values are worse at convergence (as also reflected in the first column of Table 2, and the center of mass suffers a progressive deviation from that of the native structure as the iterations progress.

Table 2: Grid search over regularization strengths $\lambda_{\mathrm{DM}}$ and $\lambda_{\mathrm{torsion}}$. Each cell reports the motif RMSD at convergence, averaged across three test cases (PDB IDs: 1PRW, 7MRX and 6HAM).

| $\lambda_{\mathrm{DM}}$ \ $\lambda_{\mathrm{torsion}}$ | 0.005 | 0.01 | 0.05 | 0.1 | 0.5 |
|:-----:|:-----:|:-----:|:-----:|:-----:|:-----:|
| 0.005 | $0.95 \pm 0.32$ | $0.99 \pm 0.29$ | $0.93 \pm 0.19$ | $0.94 \pm 0.18$ | $1.02 \pm 0.21$ |
| 0.01 | $1.82 \pm 0.40$ | $1.02 \pm 0.29$ | $0.77 \pm 0.21$ | $1.11 \pm 0.30$ | $1.04 \pm 0.22$ |
| 0.05 | $1.58 \pm 0.37$ | $0.98 \pm 0.33$ | $0.81 \pm 0.24$ | $0.96 \pm 0.27$ | $0.99 \pm 0.24$ |
| 0.1 | $1.03 \pm 0.36$ | $1.00 \pm 0.31$ | $0.78 \pm 0.28$ | $0.80 \pm 0.29$ | $0.83 \pm 0.25$ |
| 0.5 | $1.19 \pm 0.42$ | $1.15 \pm 0.34$ | $0.90 \pm 0.17$ | $0.87 \pm 0.25$ | $0.92 \pm 0.22$ |

# H  PGEL EVALUATION RESULTS

Table 3: Number of designable backbones (out of 1000) and self-consistent clusters (diversity) for PGEL and partial diffusion across benchmarking tasks.

| PDB | Designability | | Diversity | |
|---|---|---|---|---|
| | Partial diff. | PGEL | Partial diff. | PGEL |
| 1BCF | 983 | **1000** | 0 | **1** |
| 1PRW | 411 | **1000** | 0 | **2** |
| 1YCR | 252 | **802** | 1 | **6** |
| 3IXT | 393 | **502** | 1 | **3** |
| 5IUS | 434 | **711** | 0 | **2** |
| 6E6R | 523 | 523 | 0 | **13** |
| 6EXZ | **983** | 887 | 1 | **5** |
| 6HAM | 798 | **823** | 8 | **15** |
| 7MRX | 331 | **990** | 6 | **10** |
| 7OW1 | 948 | **950** | 2 | **4** |

Table 4: Detailed results of PGEL successes.

| PDB & design ID | $\alpha$ | $\omega$ | mRMSD (Å) | Motif pLDDT | Sequence | mRMSD AF3 (Å) | pAE |
|---|---|---|---|---|---|---|---|
| **1BCF** | | | | | | | |
| 699 | 0.9 | 0 | 1.99 | 76 | GLIGLGSS…RVEIALKN…EAAKEFLE…IAEEVEKR | 1.97 | 2.17 |
| **1PRW** | | | | | | | |
| 17 | 0.75 | 1 | 1.78 | 79 | FRVIAGGEDGLVTLEQLARY/VRRVAGRGGRLISFEDFLAI | 1.54 | 4.43 |
| 860 | 0.5 | 1 | 1.96 | 81 | ARWLDKGGSGAVFGEQLGEF/VAAALEGGKEAKLEEWFLNY | 1.25 | 4.84 |
| **1YCR** | | | | | | | |
| 14 | 0.3 | 1 | 1.36 | 75 | GSLEELLAQLDGG | 1.56 | 2.88 |
| 36 | 0.7 | 1 | 1.09 | 72 | MGLMELLLGVPGS | 1.25 | 3.19 |
| 285 | 0.3 | 1 | 1.59 | 72 | ASGLELLKKFKLP | 1.60 | 3.53 |
| 334 | 0.65 | 1 | 1.03 | 72 | SSLMELLEKIDIE | 1.25 | 2.88 |
| 619 | 0.2 | 1 | 0.59 | 87 | ESFEELWKKIPGS | 1.70 | 2.45 |
| 695 | 0.25 | 1 | 1.04 | 74 | SSMWELWQEIEGE | 0.86 | 2.42 |
| **3IXT** | | | | | | | |
| 112 | 0.85 | 0 | 1.23 | 75 | NLTAVTAYFNSGVTPDELRALARN | 0.97 | 3.49 |
| 118 | 0.1 | 0 | 1.03 | 74 | NEELYKEISKTPFNGEEWIKLARN | 1.74 | 4.09 |
| 542 | 0.4 | 1 | 1.14 | 70 | NAQKAKEALDSGFNGEELRKVFSN | 1.83 | 3.82 |
| **5IUS** | | | | | | | |
| 348 | 0.75 | 1 | 1.8 | 71 | YSLQVKKTLPDGTVVTVATT…SRFTVTLIDFNGNIRLLESEPA | 0.71 | 3.61 |
| 782 | 0.4 | 1 | 1.53 | 72 | PVALCKRVLPNSSEYISGQT…GPARCVYIDLVPGGSFQEGGTA | 1.67 | 4.03 |
| **6E6R** | | | | | | | |
| 16 | 0.35 | 1 | 1.90 | 79 | TEEEVLGEKKGVC | 1.03 | 2.55 |
| 95 | 0.25 | 0 | 1.85 | 73 | TEEECLKEKVGVC | 1.97 | 1.62 |
| 97 | 0.25 | 0 | 1.72 | 81 | KEEEVLKEKLKVC | 1.99 | 1.96 |
| 121 | 0.6 | 0 | 1.02 | 75 | KGWEELRKKVGLC | 0.98 | 1.88 |
| 125 | 0.05 | 0 | 1.58 | 84 | EPEKCLTEKLGVC | 1.08 | 1.68 |
| 126 | 0.15 | 0 | 1.69 | 74 | KEEECLKEKLKVC | 1.90 | 1.70 |
| 201 | 0.4 | 0 | 1.93 | 75 | TEEEILSGKYGLC | 1.35 | 1.98 |
| 205 | 0.05 | 1 | 1.62 | 80 | KGDECLKEKIGVC | 1.82 | 1.66 |
| 306 | 0.1 | 0 | 1.86 | 82 | TPEEVLTKVLGLC | 0.89 | 1.80 |
| 522 | 0.45 | 0 | 0.93 | 70 | KTFEEIKEKLGCF | 0.94 | 1.84 |

*Continued on next page*

| PDB & design ID | $\alpha$ | $\omega$ | mRMSD (Å) | Motif pLDDT | Sequence | mRMSD AF3 (Å) | pAE |
|---|---|---|---|---|---|---|---|
| 600 | 0.05 | 0 | 0.63 | 88 | KSFEEVNKKLGIT | 0.51 | 1.50 |
| 695 | 0.25 | 0 | 1.16 | 70 | RTLKCVFEKLGIL | 0.95 | 1.53 |
| 776 | 0.05 | 0 | 0.95 | 83 | KTFEEVKKKLPGL | 0.48 | 2.32 |
| **6EXZ** | | | | | | | |
| 42 | 0.75 | 0 | 1.25 | 76 | LYSLFPGGFKVTLLF | 1.35 | 3.58 |
| 166 | 0.6 | 1 | 1.11 | 85 | LSTEIKGGVEYILKL | 1.06 | 3.30 |
| 325 | 0.15 | 0 | 1.35 | 84 | LYTLAPGLLTLCIAL | 1.02 | 3.73 |
| 448 | 0.05 | 1 | 0.98 | 83 | LSELFPGGVKLVVAL | 0.89 | 4.71 |
| 699 | 0.5 | 1 | 1.82 | 74 | LWQHAPGAGGLLLVG | 1.98 | 4.56 |
| **6HAM** | | | | | | | |
| 40 | 0.45 | 1 | 1.29 | 76 | IALTRAGGLGGLIREWAEEKVG | 1.09 | 4.08 |
| 64 | 0.15 | 1 | 0.80 | 87 | ILLSRAGYLSREAARWISEKYG | 1.40 | 4.33 |
| 104 | 0.15 | 0 | 1.23 | 71 | IALTNAGWLDGLIAEFMKEKTG | 1.88 | 3.45 |
| 115 | 0.05 | 1 | 1.10 | 86 | ILLARPGSLGAEAARHLSELTG | 0.90 | 4.64 |
| 131 | 0.55 | 0 | 1.94 | 78 | IALSRGLVLDPELAQLMKELCG | 1.98 | 3.71 |
| 257 | 0.25 | 1 | 1.28 | 71 | IALERAGYRDRLIKELGKELLG | 1.40 | 4.14 |
| 313 | 0.25 | 1 | 1.78 | 71 | IALSTKGGLSGLIADFAKEVLG | 1.43 | 4.51 |
| 578 | 0.05 | 1 | 1.88 | 81 | VLLHRPGAADELARWLAEKVGG | 0.83 | 3.75 |
| 653 | 0.6 | 0 | 1.85 | 74 | ICLSRAGVFSGLFREIAEEFGK | 0.89 | 3.88 |
| 658 | 0.1 | 1 | 1.16 | 81 | IVLSRPGANGSVAREYAKEKLG | 0.79 | 4.67 |
| 669 | 0.3 | 1 | 1.75 | 72 | VFLSRNGFLHGLAREIAEELLG | 1.46 | 4.38 |
| 694 | 0.5 | 0 | 1.89 | 73 | IFLTRGGLLHGLVRELARELLK | 1.53 | 4.31 |
| 716 | 0.05 | 1 | 1.08 | 81 | IVLSRAGARSSEAGEWIKERLG | 1.22 | 4.19 |
| 839 | 0.15 | 1 | 1.28 | 81 | IALSRAGARSGEMGQIFKEITG | 1.19 | 4.30 |
| 937 | 0.4 | 1 | 1.80 | 71 | VLLSRGPYGSSLARELADYFGL | 1.49 | 4.92 |
| **7MRX** | | | | | | | |
| 0 | 0.55 | 0 | 0.74 | 79 | GLPESVSGNNQALYDSIMYDVE | 0.89 | 3.17 |
| 31 | 0.4 | 0 | 1.35 | 78 | GLPDTVKGLYAIGREAGYYYGD | 0.83 | 3.30 |
| 100 | 0.95 | 1 | 1.26 | 73 | GLPSTVTGLAGIGEDIRKGLLE | 1.78 | 3.73 |
| 114 | 0.75 | 1 | 0.83 | 75 | GASEGDAPAIYLAEDYCRYDLD | 1.22 | 3.79 |
| 145 | 0.4 | 0 | 0.78 | 78 | EFPEWVNGTLDAIYDGILYYTE | 0.69 | 4.93 |
| 308 | 0.5 | 1 | 1.57 | 74 | GIPESFKATTEAIGDWIRSNAD | 1.25 | 4.97 |
| 352 | 0.85 | 1 | 1.53 | 75 | GLPEEFTGNPYAIGEEAKRRLD | 1.98 | 3.81 |
| 730 | 0.8 | 1 | 1.85 | 72 | GIPEYLMGPDESLLDWLKSLSD | 1.42 | 4.90 |
| 744 | 0.8 | 1 | 0.88 | 75 | EVPGLEITGLSSLESTIRGYGS | 1.96 | 2.42 |
| 814 | 0.3 | 1 | 0.77 | 78 | GIKNLELSGLDAIKAAIDDLSG | 1.13 | 3.79 |
| **7OW1** | | | | | | | |
| 136 | 0.2 | 1 | 1.92 | 70 | SIDFGDGA | 1.93 | 4.92 |
| 287 | 0.05 | 1 | 0.53 | 91 | GSGGSLDT | 0.62 | 3.54 |
| 637 | 0.35 | 0 | 1.00 | 77 | GSRSGELV | 0.77 | 3.59 |
| 837 | 0.15 | 1 | 1.32 | 77 | GDNSGEAV | 0.99 | 4.26 |

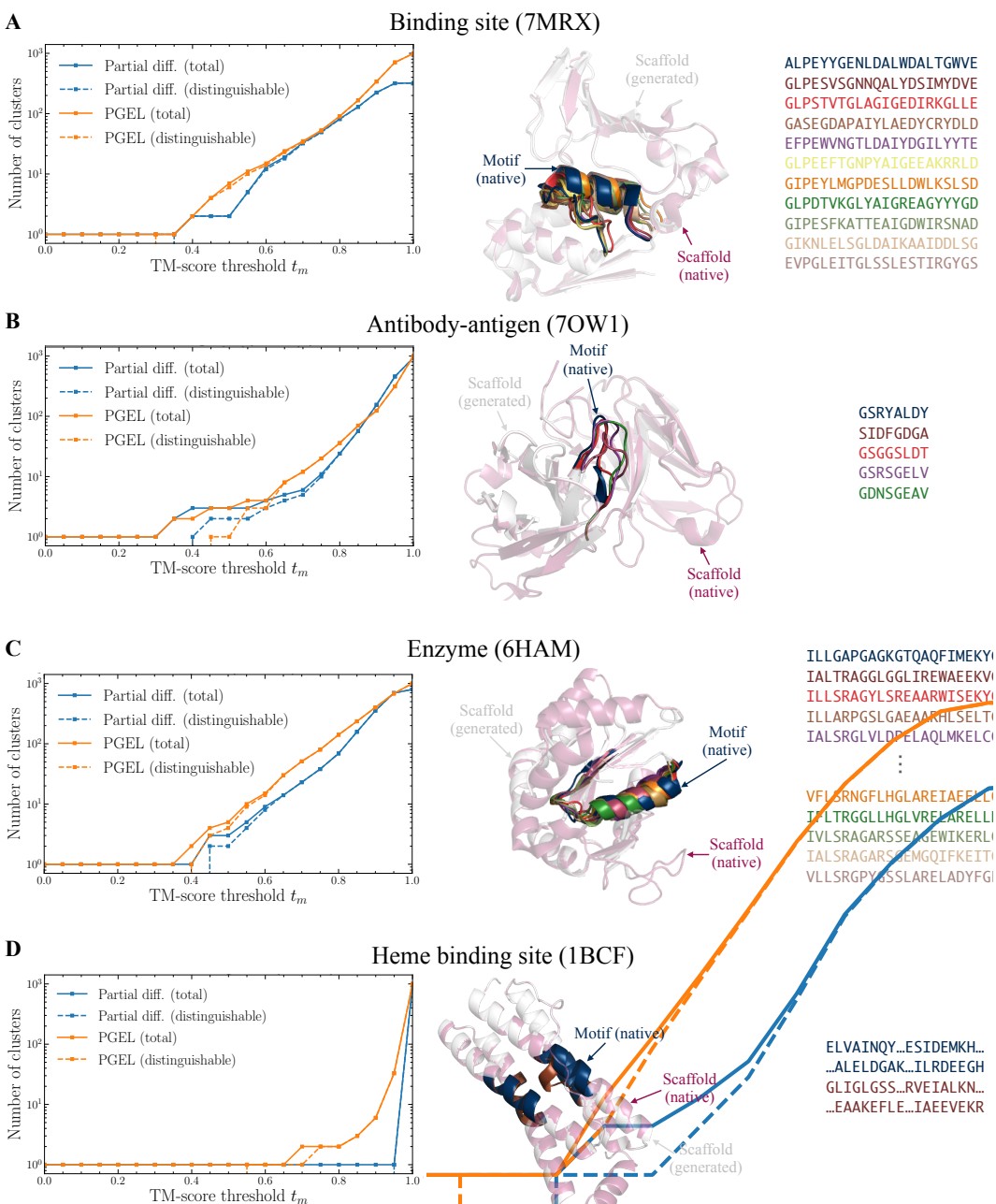

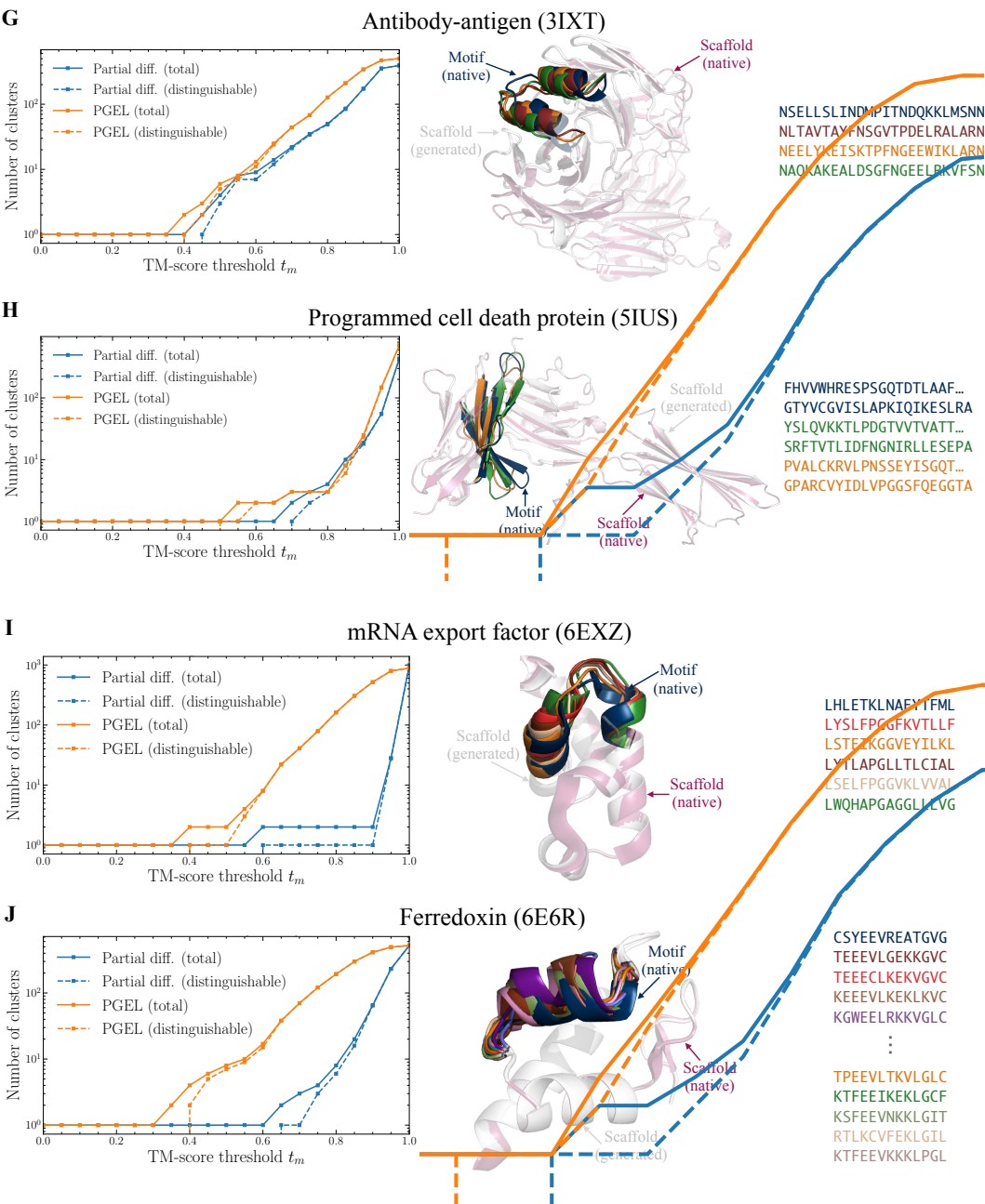

Figure 7: Same as Figure 5 but for the remaining examples. *Left:* Number of clusters identified by PGEL and partial diffusion, both total and distinguishable from native, as a function of the TM-score threshold $t_m$. *Right:* Generated binding site backbones (overlaid with native), alongside the native sequence and sequences that refold to self-consistent structures.

# I BINDING AFFINITY RESULTS

Table 5: PRODIGY-predicted binding affinities for the three examples that correspond to protein-protein complexes (PDBs 1YCR, 5IUS and 7MRX).

| PDB & design ID | Method | $\Delta G$ (kcal/mol) |
|---|---|---|
| **1YCR** | | |
| Native | – | -7.7 |
| 14 | PGEL | -7.1 |
| 36 | PGEL | -5.9 |
| 285 | PGEL | -6.8 |
| 334 | PGEL | -6.8 |
| 619 | PGEL | -6.6 |
| 695 | PGEL | -7.6 |
| 101 | Partial diff. | -6.4 |
| **5IUS** | | |
| Native | – | -9.7 |
| 348 | PGEL | -10.0 |
| 782 | PGEL | -9.7 |
| **7MRX** | | |
| Native | – | -11.4 |
| 0 | PGEL | -9.3 |
| 31 | PGEL | -11.4 |
| 100 | PGEL | -10.8 |
| 114 | PGEL | -9.5 |
| 145 | PGEL | -10.1 |
| 308 | PGEL | -10.4 |
| 352 | PGEL | -12.8 |
| 730 | PGEL | -11.6 |
| 744 | PGEL | -9.5 |
| 814 | PGEL | -9.7 |
| 1 | Partial diff. | -9.5 |
| 18 | Partial diff. | -10.9 |
| 307 | Partial diff. | -8.5 |
| 327 | Partial diff. | -8.1 |
| 513 | Partial diff. | -9.7 |
| 780 | Partial diff. | -9.4 |

