# OpenReview forum: "Protein generation with embedding learning for motif diversification"
_ICLR.cc/2026/Workshop/LMRL — ICLR 2026 Workshop LMRL Poster_

### Official Review · Reviewer_2iPy · 2026-02-22
**Review of Submission 42**

**Rating:** 8
**Confidence:** 4

**Review:**

The Protein Generation with Embedding Learning (PGEL) framework introduces a shift in protein design by moving away from traditional coordinate-space perturbations toward high-dimensional embedding space. By learning a specific motif representation within the frozen denoiser of a pre-trained diffusion model and pairing it with a novel scaffold MSA masking technique, the method solves the "diversity-fidelity" trade-off that typically plagues partial diffusion. Across 10 cases—including cancer-related complexes, antibodies, and enzymes—PGEL consistently generates backbones that are significantly more structurally diverse while maintaining higher designability and self-consistency than standard stochastic baselines.

The major strength of PGEL is its ability to explore expansive backbone manifolds while strictly anchoring the motif to the geometric constraints necessary for biological function. Due to the rich representational capacity of existing models like RFdiffusion without requiring expensive retraining, it remains a highly adaptable tool with a per-timestep inference speed nearly identical to current methods.

However, the approach adds a layer of computational friction, as the initial embedding optimization requires an additional 2 to 20 minutes of GPU time per protein system. Furthermore, while the framework excels in in silico validation—meeting rigorous pLDDT, RMSD, and AlphaFold3 refolding thresholds—these designs still lack wet-lab experimental validataion to prove their actual therapeutic or catalytic activity

---

### Official Review · Reviewer_RHZQ · 2026-02-23
**Embedding-Space Motif Diversification for Protein Diffusion via Learned Motif Embeddings and MSA Masking**

**Rating:** 7
**Confidence:** 3

**Review:**

**Summary**: This paper proposes **PGEL**, a framework that learns a **motif embedding** inside a *frozen* RFdiffusion denoiser/encoder, then increases motif diversity by **masking scaffold MSA embeddings** during generation, aiming to improve the diversity–fidelity trade-off seen in partial diffusion.

**Quality**
- The method is reasonably well-specified (embedding optimization + masking-based generation) and evaluated on **10 cases** with a consistent protocol against partial diffusion.
- The evaluation pipeline is sensible for *in silico* motif diversification (designability filters, clustering by TM-score, then self-consistency via ProteinMPNN + AlphaFold3).

**Clarity**
- The core objective is clear and the loss is explicit
- Some key implementation details remain high-level (e.g., exact definition/shape of learned embeddings across “state/pair/MSA”, and how gradients flow through the frozen blocks), which may hinder reproducibility without code.

**Originality**
- The main conceptual novelty is adapting **textual inversion** to protein diffusion by learning a **motif embedding** and perturbing **in embedding space** rather than coordinate space.
- The masking strategy on scaffold MSA embeddings is a simple but interesting control knob for relaxing constraints; it’s plausible and empirically helpful, though somewhat heuristic.

**Significance**
- If robust, this could be a practical add-on to existing diffusion backbones (no retraining of the denoiser) for applications needing “controlled” motif variation (binders/enzymes).
- However, the paper’s claims are currently limited by reliance on predicted metrics (pLDDT, AF3 self-consistency, PRODIGY ΔG) without experimental validation.

### Strengths
- **Clear formulation of the task** (motif diversification with fixed scaffold in real space) and well-motivated comparison to partial diffusion’s diversity–fidelity issue.
- **Concrete method** with explicit losses (coordinate MSE, Cα distance matrix, torsion) that encode both local and global motif geometry.
- **Empirical gains** reported in designability and diversity (self-consistent clusters) on most of the 10 cases under the proposed metrics.
- **Computationally comparable generation cost** to partial diffusion per timestep, with an added embedding-optimization overhead that is reported and bounded.

### Weaknesses
- **Masking is under-justified theoretically**: zeroing rows/columns of MSA embeddings is a strong intervention, and it’s unclear when it preserves functional constraints versus creating artifacts.
- **Evaluation depends on thresholds** (e.g., pLDDT≥70, scRMSD≤1Å, mRMSD≤2Å, tm=0.6), and it’s unclear how sensitive conclusions are to these choices across tasks.
- **Binding affinity results are mixed** (e.g., in p53-MDM2, many PGEL clusters have weaker predicted affinity), which complicates the “viable diversification” narrative.

---

### Official Review · Reviewer_cw7c · 2026-02-25
**Adapting Textual Inversion to Motif Diversification in Protein Design**

**Rating:** 8
**Confidence:** 4

**Review:**

### **Summary**

This paper introduces Protein Generation with Embedding Learning (PGEL), a framework for motif diversification that builds on a frozen RFdiffusion denoiser and adapts the idea of textual inversion from latent diffusion models in computer vision. Rather than perturbing atomic coordinates directly (as in partial diffusion), PGEL learns a high-dimensional motif embedding and performs controlled perturbations in embedding space. The goal is to generate motif backbones that are structurally diverse yet still compatible with biological function.

The motif embedding is optimized while the RFdiffusion encoder and denoiser remain frozen. The scaffold sequence and structure are encoded once; the learned motif embedding and scaffold embeddings jointly condition the denoiser. The loss combines backbone MSE, an alpha-carbon distance-matrix term (rigid-motion invariant), and torsion-angle penalties. Generation uses the optimized motif embedding together with scaffold MSA embeddings, which are perturbed via column-wise masking to relax geometric constraints. Ablations suggest that both the learned motif embedding and MSA masking are important for achieving diversity while maintaining high pLDDT in the motif.

Across ten case studies, PGEL outperforms RFdiffusion’s partial diffusion baseline in structural diversity (TM-score clustering), designability (motif pLDDT ≥ 70, scaffold RMSD ≤ 1Å, motif RMSD ≤ 2Å), and self-consistency (at least one of 8 ProteinMPNN-designed sequences refolds in AlphaFold3 with motif RMSD ≤ 2Å and pAE ≤ 5). Overall, I found the approach well-motivated and clearly explained. It leverages pretrained components instead of retraining a generative model from scratch, and the conceptual bridge to textual inversion is innovative. My main concerns relate to biological interpretation, evaluation depth, and scalability of the per-target embedding optimization.

### **Strengths and Weaknesses**

### **Strengths:**
- Clear motivation: the paper precisely identifies the diversity–fidelity tradeoff in partial diffusion and proposes a principled alternative, inspired by textual inversion.
- Computational efficiency: PGEL reuses frozen RFdiffusion components, avoiding expensive retraining.
- Strong empirical comparison to the prevailing baseline (partial diffusion), with consistent improvements in diversity and designability across most test cases.
- The inverted evaluation pipeline (filter for diversity/designability before running expensive self-consistency checks) is practical and well justified.
- Transparent reporting of runtime (generation speed ~0.71s per timestep, comparable to partial diffusion) and embedding optimization cost (2–20 minutes per target).
- The authors clearly acknowledge limitations and outline concrete future directions.

### **Weaknesses:**
- The biological rationale for fixing the scaffold rigidly while diversifying the motif is not fully justified. It would help to articulate scenarios where rigid scaffold preservation is essential, rather than simply convenient.
- The p53-MDM2 example is compelling but effectively resembles binder redesign. Introducing this application earlier in the paper would strengthen the motivation.
- Beyond the first paragraph of Section 3.1, the mathematical connection between PGEL and textual inversion is not fully developed. A more explicit derivation (perhaps in the Appendix) would solidify the conceptual contribution.
- Diversity is measured primarily via TM-score clustering under tight RMSD constraints. It is unclear how much meaningful conformational variation can exist while enforcing motif RMSD ≤ 2Å.
- The evaluation criteria (designability, diversity, self-consistency) are standard and appropriate, but not especially creative. They remain several steps removed from true biological function.
- The pLDDT ≥  70 constraint for the motif is sensible for structure-based design, but it limits PGEL’s ability to design intrinsically disordered, yet functional, motifs. In the p53-MDM2 case, the original motif is intrinsically disordered before engaging with MDM2. The novel, designed motif that would effectively perform this function could be intrinsically disordered out of complex, and AlphaFold may struggle to predict its transition. The paper would be strengthened by a consideration of how intrinsic disorder can be factored into the design process.
- The motif embedding must be learned independently for each target structure. This adds overhead and may limit scalability. Is it possible to create a generalized PGEL that can take any PDB structure as input without training a new embedding?
- Limitations explicitly addressed by authors: motif size is constrained (practical limit ~50 residues), limiting applicability to larger functional regions. All validation is computational; no experimental verification.

### **Questions for the Authors:**
- What is the biological motivation for strictly fixing the scaffold during diversification? In what design scenarios is this constraint essential?
- How sensitive are results to the mRMSD < 2A threshold, and how was this threshold selected? Would slightly looser thresholds reveal more meaningful diversity?
- Why does eliminating MSA masking produce essentially no diversity, given that masking is applied only to scaffold embeddings, and RFdiffusion’s stochasticity should generate diversified motifs (as mentioned in Appendix D)?
- Could the learned motif embedding generalize across related motifs, or must optimization be repeated independently for every new PDB structure?
- Could diversity be evaluated in function-aware space?
- How would the method behave on intrinsically disordered motifs or regions that only adopt structure upon binding? The p53-MDM2 example is promising and indicates that this important application should be explored more.

### **Final Assessment**

This paper presents a well-defined and thoughtfully executed method for motif diversification. Its key contribution is conceptual: reframing motif variation as embedding optimization within a pretrained diffusion model, inspired by textual inversion. The approach is computationally efficient, empirically strong relative to the baseline, and theoretically interesting. While some evaluation choices and biological motivations could be more deeply justified, the work represents a meaningful and creative step forward in controlled protein design.

---

### Meta-Review · Area_Chair_MXdS · 2026-02-27

**Recommendation:** Accept (Poster)
**Confidence:** 5

**Metareview:**

Accept.

---

### Decision · Program_Chairs · 2026-03-02

**Decision:**

Accept (Spotlight)

**Comment:**

Please see the meta-review.